# Synergizing Dynamic Score Aggregation with Contrastive Regularization for Open-Set Semi-Supervised Out-of-Distribution Detection

## Abstract

Semi-supervised learning (SSL) has achieved remarkable progress by leveraging both limited labeled data and abundant unlabeled data. However, unlabeled datasets often contain out-of-distribution (OOD) samples from unknown classes, which can lead to performance degradation in open-set SSL scenarios. Current approaches primarily address this issue by identifying outliers through OOD detection. Yet, methods relying solely on neural networks are constrained by the absence of labeled OOD samples for supervision. To overcome this limitation, we propose a novel open-set OOD detection framework named **SDM**, which **S**ynergizes **D**ynamic Score Aggregation (DSA) and **M**atrix Contrastive Regularization (MCR). Specifically, we formulate OOD detection as a semi-unbalanced optimal transport (SemiUOT) problem and derive pseudo-labels by solving it. The DSA module dynamically converts SemiUOT into a classical optimal transport (OT) formulation. Unlike existing OT-based methods, DSA provides theoretically grounded and more accurate pseudo OOD scores while avoiding the direct computation of the transport plan. Meanwhile, the MCR module enhances feature discrimination through contrastive learning, thereby improving overall performance. Empirical results demonstrate the superiority of SDM. Additionally, we conduct extensive analytical experiments to elucidate the properties of each component.

## 1 Introduction

Semi-supervised learning (SSL) is a pivotal machine learning paradigm that leverages abundant unlabeled data alongside limited labeled data (Xiao et al., 2024; Berthelot et al., 2019; Zheng et al., 2022; Kingma et al., 2014; Chen et al., 2024; Yang et al., 2024; Min et al., 2024). Traditional SSL relies on the assumption that both labeled and unlabeled data share the same class space and distribution (Li et al., 2023). However, this closed-world setting is often unrealistic. In real-world open-set SSL scenarios, unlabeled data frequently contain out-of-distribution (OOD) samples from unknown classes, which can severely degrade model performance if not properly handled.

To alleviate this problem, researchers begin to explore the identification of outliers (Qin et al., 2024; Shen et al., 2024; Kaushik et al., 2024), namely OOD detection. While one possible solution is to directly implement OOD detection during testing (Hendrycks & Gimpel, 2016; Liu et al., 2020; Huang et al., 2021b; He et al., 2022; Ma et al., 2023), this approach only considers the testing phase with the model trained on purely in-distribution (ID) data. In open-set SSL, the model must be jointly optimized on both labeled and unlabeled data while ensuring the OOD detection module learns effectively without compromising classification accuracy. Consequently, (Yu et al., 2020b; Guo et al., 2020) propose to integrate OOD detection with semi-supervised learning during the training phase. However. the absence of reliable OOD labels hinders effective learning and the joint optimization framework risks sacrificing closed-set classification accuracy.

To address this challenge, recent works (Saito et al., 2021; Ren et al., 2024) introduce third-party proxies into the training phase. For instance, (Saito et al., 2021) employs a one-vs-all (OVA) classifier for OOD detection. However, this method requires training a separate classifier for each known class, which is computationally inefficient. Although (Ren et al., 2024) proposes to train a neural binary classifier using pseudo OOD scores obtained through entropy-regularized optimal transport (OT), the

entropy regularization can compromise label accuracy, and the practice of artificially amplifying the weights of unlabeled samples lacks a theoretical foundation.

To resolve this dilemma, we model the OOD detection task as a semi-unbalanced optimal transport (SemiUOT) problem and propose a **dynamic score aggregation** (DSA) module. DSA dynamically converts the SemiUOT problem into classical OT, eliminating the need for entropy regularization or ad-hoc weight amplification. This allows it to produce more accurate pseudo OOD scores for supervising the neural binary classifier. Furthermore, we introduce a **matrix contrastive regularization** (MCR) module, which enhances feature discrimination through contrastive learning without incurring inference-time overhead. Integrating these components, we propose **SDM**, a novel open-set semi-supervised OOD detection framework that **S**ynergizes **D**SA and **M**CR. Extensive experiments demonstrate the superiority of SDM over existing methods.

The main contributions are listed as follows:

- We innovatively propose the DSA module, which dynamically converts the SemiUOT problem into classical OT. This theoretically grounded approach yields more accurate pseudo-labels efficiently, overcoming key limitations of existing OT based methods. Furthermore, we introduce a fast approximation algorithm for DSA with a theoretically analyzed error bound.

- We design the MCR module as an auxiliary task based on self-supervised learning. MCR enhances feature discrimination without introducing inference latency. Integrating DSA and MCR, we construct SDM, a novel framework that synergistically combines OT with self-supervised regularization for open-set semi-supervised OOD detection.

- Extensive experiments on benchmark datasets demonstrate that SDM achieves competitive, and often superior, performance compared to state-of-the-art methods. Additional ablation and analytical studies provide insights into the properties and individual contributions of each component.

## 2 RELATED WORKS

### 2.1 OPTIMAL TRANSPORT

OT quantifies the discrepancy or distance between two distributions by calculating the minimum transport cost (Khamis et al., 2024). As a mathematical tool, recent advances have shown the promising potential of OT for various machine learning tasks, such as natural language processing (Cheng et al., 2024a; Sun et al., 2023; Cheng et al., 2024b), computer vision (Li et al., 2025; Lin & Chan, 2023; Izquierdo & Civera, 2024; Chowdhury et al.), graph matching and representation (Maretic et al., 2022; Zeng et al., 2023; 2024), generative models (Tong et al., 2023a;b; Li et al., 2024; Hui et al., 2025; Choi et al., 2023), and reinforcement learning (Klink et al., 2024; Asadulaev et al., 2024; Sun et al., 2025). Classical OT is the most basic form of OT theory, where both marginal equality constraints on source and target distributions are preserved.

**Definition 2.1** (Classical OT). *Given two distributions $\boldsymbol{\alpha}$ and $\boldsymbol{\beta}$, each containing $M$ and $N$ units, $\boldsymbol{a}$ and $\boldsymbol{b}$ are the weights vectors of them, and $\boldsymbol{C} \in \mathbb{R}_+^{M \times N}$ is the transport cost matrix. The classical OT is to find the optimal transport plan $\boldsymbol{\pi} \in \mathbb{R}_+^{M \times N}$ in the feasible solution set $\Pi$.*

$$\min_{\boldsymbol{\pi} \in \Pi(\boldsymbol{\alpha}, \boldsymbol{\beta})} \langle \mathbf{C}, \boldsymbol{\pi} \rangle_F, \Pi(\boldsymbol{\alpha}, \boldsymbol{\beta}) := \left\{ \boldsymbol{\pi} \in \mathbb{R}_+^{M \times N} : \boldsymbol{\pi} \mathbf{1}_N = \boldsymbol{a}, \boldsymbol{\pi}^{\mathrm{T}} \mathbf{1}_M = \boldsymbol{b} \right\}. \quad (1)$$

Since classical OT is essentially a linear optimization process, it can be solved exactly using any linear solver. However, when trying to make OT work in the real world, things are often trivial. If one of the two marginal constraints is relaxed, SemiUOT is defined as follows.

**Definition 2.2** (SemiUOT with KL-Divergence). *SemiUOT is to find the optimal transport plan when the target constraint on the source distribution is removed (Le et al., 2021), the formulation is,*

$$\min_{\boldsymbol{\pi} \in \Pi(\boldsymbol{\alpha}, \boldsymbol{\beta})} \langle \mathbf{C}, \boldsymbol{\pi} \rangle_F + \tau_a KL(\boldsymbol{\pi} \mathbf{1}_N \| \boldsymbol{a}), \Pi^s(\boldsymbol{\alpha}, \boldsymbol{\beta}) := \left\{ \boldsymbol{\pi} \in \mathbb{R}_+^{M \times N} : \boldsymbol{\pi}^{\mathrm{T}} \mathbf{1}_M = \boldsymbol{b} \right\}, \quad (2)$$

*where $\tau_a$ is the weight parameter. The divergence term KL is used to measure the discrepancy between the marginal distribution of the transport plan and the source distribution.*

A common approach to SemiUOT involves introducing an entropy regularization term and solving the resulting objective with a Sinkhorn-like algorithm (Cuturi, 2013), which may compromise accuracy.

## 2.2 RELATIONS BETWEEN OOD DETECTION AND OPTIMAL TRANSPORT

The motivation for the OT-based OOD detection is intuitive. For the probability distribution of both the source and the target domain (e.g., the feature embedding of unlabeled and labeled data), the common assumption is that they both obey the uniform discrete distribution. If we model OOD detection problem as classical OT on this basis, it is difficult to distinguish ID samples from OOD samples via the transport plan $\boldsymbol{\pi}$. Thus, (Ren et al., 2024) enlarges the weights of the unlabeled data to force the marginal constraint of target domain (i.e., $\boldsymbol{\pi}^{\mathrm{T}}\mathbf{1}_N = \boldsymbol{a}$) to be relaxed. The probability that the sample with less total transmission quality is an OOD sample is greater, and vice versa. Then, the pseudo OOD score obtained via the OT module can be computed as, $\mathbf{S} = \frac{1}{k}\boldsymbol{\pi}\mathbf{1}_N$, assuming that the target domain is the feature embedding of unlabled data. Due to the mechanism above, it is unrealistic to deploy it directly during testing. A more practical approach is to use $\mathbf{S}$ as the labels to train the neural OOD detector. However, the redundant mass $k$ based OOD detection lacks theoretical support and the common Sinkhorn-like OT solvers risk producing unreliable pseudo OOD score. In this paper, we directly model the OOD detection task as SemiUOT and tackle it in a more accurate way.

## 2.3 SELF-SUPERVISED REGULARIZATION FOR SEMI-SUPERVISED LEARNING

Common SSL strategies are pseudo-labeling and consistency regularization (Sohn et al., 2020; Li et al., 2022b;a; Sosea & Caragea, 2023; Ihler et al., 2024). Slightly different from SSL, the data for self-supervised learning is all unlabeled (Grill et al., 2020; Zbontar et al., 2021; Zhang et al., 2023; Lu et al., 2024; Zhang et al., 2024). Interestingly, when self-supervised learning is used as an auxiliary task for representation learning, performance improvement of semi-supervised learning can be observed (Zhai et al., 2019). For example, (Lee et al., 2022) addresses the inefficiency of traditional consistency regularization strategies in SSL by leveraging the contrastive learning mechanism of unlabeled data. (Li et al., 2021) proposes contrastive graph regularization to jointly optimize class probabilities and low-dimensional embeddings. Whether such strategy is effective in OOD tasks is still an underexplored research topic. Based on contrastive learning, we design the MCR module to further boost the performance of the model.

# 3 METHODOLOGY

## 3.1 PROBLEM DEFINITION

**Source and target domain sample:** The source domain sample $\mathbf{X}_u \in \mathbb{R}^{M \times H \times W \times CH}$ is composed of unlabeled data with known classes and unlabeled data with unknown classes (i.e., OOD sample), where $M$ is the batch size of the source domain sample and $H, W, CH$ are the height, width and number of channels of the input image date respectively. The target domain sample is the labeled data $\mathbf{X}_l \in \mathbb{R}^{N \times H \times W \times CH}$ with known classes, where $N$ is the batch size of the target domain sample. Both the labeled sample and the unlabeled sample with known classes are called ID sample. Note that $M$ should be larger than $N$ to comply with the setting of open-set SSL.

**Open-set semi-supervised OOD detection:** Given $\mathbf{X}_u$ and $\mathbf{X}_l$, the open-set semi-supervised OOD detection is to detect OOD samples in $\mathbf{X}_u$ with $\mathbf{X}_l$ as a reference and perform semi-supervised learning over both $\mathbf{X}_u$ and $\mathbf{X}_l$. This paper focuses on training neural networks to predict the OOD probability $\hat{\mathbf{S}} \in \mathbb{R}_+^{M+N}$ under the supervision of pseudo labels obtained by OT and preserving the closed-set classification accuracy through self-supervised regularization.

## 3.2 OVERVIEW

As shown in Figure 1, SDM is composed of three modules, namely the semi-supervised learning module, the OOD detection module, and the MCR module. The closed-set classification module $g$ adopts the same architecture as (Sohn et al., 2020; Ren et al., 2024). For a batch of sample from $\mathbf{X}_u$ and $\mathbf{X}_l$, we perform weak augmentation once and strong augmentation twice on $\mathbf{X}_u$ to obtain $\mathbf{X}_w, \mathbf{X}_s^1, \mathbf{X}_s^2 \in \mathbb{R}^{M \times H \times W \times CH}$. The specific augmentation method is consistent with (Sohn et al., 2020). The augmented source and target sample are mapped to their feature embedding $\mathbf{Z} = f(\mathbf{X}) \in \mathbb{R}^d$ through feature encoder $f$, where $d$ is dimension of feature embedding. The pseudo

Figure 1: The left part of the figure depicts the overall framework of SDM and the right part of the figure shows the details of the MCR (top) and DSA (bottom) module."Tied" indicates weight sharing.

OOD scores $\mathbf{S}$ of the unlabeled samples are estimated by the DSA module and OOD loss is,

$$\mathcal{L}_{ood} = \frac{1}{M+N}\|\hat{\mathbf{S}} - \mathbf{S}'\|_2^2, \tag{3}$$

where $\hat{\mathbf{S}} = h(\mathbf{X}_l \oplus \mathbf{X}_w)$, $\mathbf{S}' = \mathbf{S} \oplus \mathbf{1}_N$ and $\oplus$ indicates the concat operation. By computing the weighted sum of alignment loss and uniformity loss between $\mathbf{X}_s^1$ and $\mathbf{X}_s^2$ as $\mathcal{L}_{mcr}$, the MCR module works as an auxiliary task to enforce the feature encoder to learn better embedding. Putting all of these together according to weight, the total loss is,

$$\mathcal{L} = \mathcal{L}_x + \mathcal{L}_u + \gamma_1 \mathcal{L}_{ood} + \gamma_2 \mathcal{L}_{mcr}, \tag{4}$$

where $\mathcal{L}_x$ and $\mathcal{L}_u$ are supervised loss and unsupervised loss in Fixmatch (Sohn et al., 2020), whose weight coefficients are $\gamma_1$ and $\gamma_2$ respectively. The detailed algorithm of SDM is shown in Section A.1.

### 3.3 DYNAMIC SCORE AGGREGATION

We propose the DSA approach to estimate pseudo OOD scores in a more accurate and efficient manner, supported by theoretical guarantees. To further accelerate the computation, we introduce a targeted approximation method. Theoretical analysis also establishes an approximation error bound.

Firstly, We adopt the formulation of SemiUOT as Equation (2) to model the OOD detection task as the SemiOT problem, where $\boldsymbol{\alpha} = \sum_{i=1}^M \frac{1}{M}\delta_{\mathbf{u}_i}, \boldsymbol{\beta} = \sum_{i=1}^N \frac{1}{N}\delta_{\mathbf{l}_i}$ are the distribution weight of source samples and target samples respectively, and $\delta$ is the Dirac function. Given the feature embedding of labeled sample $\mathbf{Z}_l$ and that of strong augmented unlabeled sample $\mathbf{Z}_s^1$, we calculate the transport cost from $\mathbf{Z}_s^1$ to $\mathbf{Z}_l$ as $\mathbf{C} = \mathbf{1}_{M\times N} - \frac{\mathbf{Z}_s^1\mathbf{Z}_l}{\|\mathbf{Z}_s^1\|_2 \cdot \|\mathbf{Z}_l\|_2}$. Then, instead of directly solving the SemiUOT problem via Sinkhorn-like approach, we have the following proposition to transform the SemiUOT problem to classical OT and the problem can be tackled in a more accurate and efficient way.

**Proposition 3.1** *(Dynamic Score Aggregation with Exact SemiUOT Solver). Given SemiUOT with KL-Divergence shown in Equation* (2)*, we can rewrite its dual form as below:*

$$\min_{\boldsymbol{u},\boldsymbol{v},\boldsymbol{s},\zeta} \mathcal{J} = \tau_a \left\langle \boldsymbol{a}, \exp\left(-\frac{\boldsymbol{u}+\zeta}{\tau_a}\right) \right\rangle - \langle \boldsymbol{v} - \zeta, \boldsymbol{b}\rangle, s.t. u_i + v_j + s_{ij} = C_{ij}, s_{ij} \geq 0, \tag{5}$$

*where $\boldsymbol{u}$, $\boldsymbol{v}$, $s$, and $\zeta$ are dual variables. Equation* (5) *can be rewritten as classical OT:*

$$\min_{\boldsymbol{\pi} \geq 0} \mathcal{J}_{\mathrm{S}} = \langle \boldsymbol{C}, \boldsymbol{\pi}\rangle, s.t. \boldsymbol{\pi}\mathbf{1}_N = \boldsymbol{a} \odot \exp\left(-\frac{\boldsymbol{u}^* + \zeta^*}{\tau_a}\right), \boldsymbol{\pi}^{\mathrm{T}}\mathbf{1}_M = \boldsymbol{b}, \tag{6}$$

where $\boldsymbol{u}^*$ and $\zeta^*$ are the optimal value of $\boldsymbol{u}$ and $\zeta$. The detailed proof is in Section A.3. That is, if we partially solve $\boldsymbol{u}$ and $\zeta$, SemiUOT can be simplified to classical OT. Common exact solutions for $\boldsymbol{u}$

and $\zeta$ are L-BFGS (Liu & Nocedal, 1989) or FISTA (Beck & Teboulle, 2009). The key insight is that solving the simplified classical OT for the OOD score using an exact OT solver is equivalent to obtaining it directly from the source weight in Equation (6).

Since the above approach is to obtain the equivalent form of SemiUOT via dynamically reweighting the weights of the source distribution, we name it **dynamic score aggregation** (DSA). By eschewing the entropy regularization responsible for unreliable scores and the arbitrary amplification of source weights, DSA achieves more accurate OOD detection without the computational burden of directly solving the full optimal transport plan. However, if we review the exact SemiUOT equation:

$$\min_{\boldsymbol{u}, \zeta} \mathcal{J}_{\mathrm{S}} = \tau_a \sum_{i=1}^{M} a_i \exp\left(-\frac{u_i + \zeta}{\tau_a}\right) - \sum_{j=1}^{N}\left[\inf_{k \in [M]}[C_{kj} - u_k] - \zeta\right] b_j, \qquad (7)$$

although $\mathcal{J}_{\mathrm{S}}$ is convex and has unique solutions, the presence of $\inf(\cdot)$ renders it a non-smooth function, leading to inefficient optimization. Thus, to further accelerate the optimization process, we turn to the approximate solution of DSA with the following proposition.

**Proposition 3.2** *(Accelerating DSA with approximation). We consider a smooth approximation to replace* $\inf(\cdot)$ *as* $\inf_{k \in [M]}[C_{kj} - f_k] \approx -\epsilon \log[\sum_{k=1}^{M} e^{\frac{f_k - C_{kj}}{\epsilon}}]$. *Note that* $\epsilon > 0$ *denotes the balanced hyperparameter among precision and smoothness of the function. Smaller* $\epsilon$ *(e.g.,* $\epsilon$ *approaches to 0) could lead to more accurate while less smooth solutions. Then we can obtain the proposed Approximate SemiUOT Equation as* $\widehat{\mathcal{J}}_{\mathrm{S}}$ *by replacing* $\inf(\cdot)$ *with the smoothness term for* $\widehat{f}$,

$$\min_{\boldsymbol{u}, \zeta} \widehat{\mathcal{J}}_{\mathrm{S}} = \tau_a \exp\left(-\frac{\zeta}{\tau_a}\right) \sum_{i=1}^{M} a_i \exp\left(-\frac{u_i}{\tau_a}\right) + \sum_{j=1}^{N}\left[\epsilon \log\left[\sum_{k=1}^{M} \exp\left(\frac{u_k - C_{kj}}{\epsilon}\right)\right] + \zeta\right] b_j. \quad (8)$$

*Then,* $u_i$ *can be iteratively updated as follows:*

$$u_i^{(l+1)} = \frac{\tau_a \epsilon}{\tau_a + \epsilon} \log\left(a_i \exp\left(-\frac{\zeta}{\tau_a}\right)\right) - \frac{\tau_a \epsilon}{\tau_a + \epsilon} \log\left[\sum_{j=1}^{N}\left[\frac{\exp\left(-\frac{C_{ij}}{\epsilon}\right)}{\sum_{k=1}^{M} \exp\left(\frac{u_k^{(l)} - C_{kj}}{\epsilon}\right)}\right] b_j\right] \quad (9)$$

$$= \mathcal{T}(u_i^{(l)}).$$

*Meanwhile,* $\zeta$ *can be computed as,*

$$\zeta = \tau_a \left[\log\left(\sum_{i=1}^{M} a_i \exp\left(-\frac{u_i}{\tau_a}\right)\right) - \log\left(\sum_{j=1}^{N} b_j\right)\right]. \qquad (10)$$

With this approximation method, we can solve DSA more efficiently. The proof is in Section A.4. As shown by the following proposition, the approximation error will become smaller with smaller $\epsilon$.

**Proposition 3.3** *(Approximation error). We consider the analysis between optimal results of* $u^o$ *and* $\hat{u}^o$ *and thus we set* $\zeta = 0$ *in SemiUOT. Then we define* $E_p(u) = \mathcal{J}_{\mathrm{S}}$ *and* $K_p(\hat{u}) = \widehat{\mathcal{J}}_{\mathrm{S}} - \epsilon \log M$. *Hence we have the following relationships: (1)* $K_P(u^o) \leq E_P(u^o) \leq E_P(\hat{u}^o) \leq K_P(\hat{u}^o) + \epsilon \log M$, *(2)* $K_P(\hat{u}^o) \leq K_P(u^o) \leq E_P(u^o) \leq K_P(u^o) + \epsilon \log M$, *and thus shows* $|K_P(u^o) - K_P(\hat{u}^o)| \leq \epsilon \log M$. *Moreover we have:*

$$|E_P(u^o) - K_P(\hat{u}^o)| \leq |E_P(u^o) - K_P(u^o)| + |K_P(u^o) - K_P(\hat{u}^o)| \leq 2\epsilon \log M. \qquad (11)$$

*Therefore we can observe that* $u$ *and* $\hat{u}^o$ *will get closer with smaller* $\epsilon$.

---

**Algorithm 1** Dynamic Score Aggregation for OOD Detection

---

**Input** : Optimization objective $\mathcal{J}_\mathrm{S}$.
**Output** : Pseudo OOD score $\mathbf{S}'$.
Initialize $t = 0$, $u^0 = (\frac{1}{M}, \frac{1}{M}, \cdots, \frac{1}{M})$ and $\zeta = 0$;
**for** $t = 0, 1, 2, \cdots, T$ **do**
$\quad$ Obtain the optimal solution on $\boldsymbol{u}^{(t+1)}$ via Equation (9).
$\quad$ Optimize $\zeta$ by considering $\frac{\partial \mathcal{J}_\mathrm{S}}{\partial \zeta} = 0$ via Equation (10).
**end**
Obtain the classical OT format $\widehat{\mathcal{J}}_\mathrm{S}$ via Equation (6).
The pseudo OOD score $\mathbf{S}$ of the unlabeled samples is computed as $\mathbf{S} = \boldsymbol{a} \odot \exp\left(-\frac{\boldsymbol{u}^* + \zeta^*}{\tau_a}\right)$.
The pseudo OOD score of the labeled samples is set as $\mathbf{1}_N$.
**return** $\mathbf{S}' = \mathbf{S} \oplus \mathbf{1}_N$.

---

The detailed computation process of the pseudo OOD scores based on approximation accelerated DSA is provided in Algorithm 1, whose time complexity is $O(MN)$. It is worth highlighting that the DSA approach can directly obtain the OOD score from the source marginal distribution of $\widehat{\mathcal{J}}_\mathrm{S}$, thus avoiding the solution of $\boldsymbol{\pi}$. With the supervision of $\mathbf{S}'$, we train the OOD detection head $h$ to distinguish the OOD samples from the ID ones.

### 3.4 MATRIX CONTRASTIVE REGULARIZATION

To improve the overall performance of SDM, we leverage the advantage of contrastive learning and propose the MCR module as an auxiliary task under the semi-supervised learning framework.

Given the feature embedding of the source domain sample from different strong augmented views, $\mathbf{Z}_s^1$ and $\mathbf{Z}_s^2$, their corresponding reconstructed feature embedding are denoted as $\mathcal{Z}_1, \mathcal{Z}_2 = \mathcal{G}(\mathbf{Z}_s^1), \mathcal{G}(\mathbf{Z}_s^2)$, where $\mathcal{Z}_1, \mathcal{Z}_2 \in \mathbb{R}^{M \times d}$ and $\mathcal{G}$ is the self-supervised learning head. We introduce the matrix information theory (Zhang et al., 2023) into open-set SSL and leverage the alignment loss and uniformity loss to construct the regularization term, guiding the feature encoder $f$ to learn more effective feature embedding and improve the performance of the other two branches,

$$\mathcal{L}_{\mathrm{align}}(\mathcal{Z}_1, \mathcal{Z}_2) = -\operatorname{tr}\left(\frac{1}{B}\mathcal{Z}_1 \mathbf{H}_B \mathcal{Z}_2^\top\right) + \gamma \cdot \mathrm{MCE}\left(\frac{1}{B}\mathcal{Z}_1 \mathbf{H}_B \mathcal{Z}_1^\top, \frac{1}{B}\mathcal{Z}_2 \mathbf{H}_B \mathcal{Z}_2^\top\right),$$

$$\mathcal{L}_{\mathrm{uni}}(\mathcal{Z}_1, \mathcal{Z}_2) = \mathrm{MCE}\left(\frac{1}{d}\mathbf{I}_d, \frac{1}{B}\mathcal{Z}_1 \mathbf{H}_B \mathcal{Z}_2^\top\right),$$

(12)

where $\mathbf{H}_B = \mathbf{I}_B - \frac{1}{B}\mathbf{1}_\mathbf{B}\mathbf{1}_\mathbf{B}^\top$. The computation of the MCE function is as follows:

$$\mathrm{MCE}(\mathbf{P}, \mathbf{Q}) = \operatorname{tr}(-\mathbf{P}\log\mathbf{Q} + \mathbf{Q}),$$

(13)

where $\mathbf{P}, \mathbf{Q}$ are two positive semi-definite matrices. The alignment loss aims to bring the positive sample pairs closer in the feature domain, while the uniformity loss forces the representation to be evenly distributed in the feature space, avoiding the aggregation of features in certain areas or even NC. As shown in Equation (12), $\mathcal{L}_{\mathrm{uni}}$ tends to guide the centered sample covariance matrix close to the unit matrix in terms of expression. That is, if $\mathcal{L}_{\mathrm{uni}}$ is minimized as much as possible, different sample categories will be more evenly distributed in the feature space. Then, the contrastive regularization term is,

$$\mathcal{L}_{mcr} = \mathcal{L}_{\mathrm{align}}(\mathcal{Z}_1, \mathcal{Z}_2) + \mathcal{L}_{\mathrm{uni}}(\mathcal{Z}_1, \mathcal{Z}_2).$$

(14)

The detailed algorithm is presented in Section A.2. With both MCR and DSA, a novel open-set OOD detection framework SDM is constructed, which trains a neural OOD detector with the OOD scores produced by DSA as the supervision and boosts the overall performance via the MCR module.

## 4 EXPERIMENTS

### 4.1 SETUP

**Datasets.** We build open-set OOD benchmarks based on CIFAR-10 and CIFAR-100 (Krizhevsky et al., 2009) with different settings. By artificially dividing known classes as labeled samples, the

Table 1: Top-1 accuracy and AUROC on CIFAR-10 benchmarks under different settings.To verify the statistical significance of the results, we report both the mean and standard deviation of 5 experiments.

| # of Labeled Classes | 50 | | 100 | | 400 | |
|---|---|---|---|---|---|---|
| Metric | Acc | AUROC | Acc | AUROC | Acc | AUROC |
| MTCF | 79.7±0.9 | 96.6±0.5 | 86.3±0.9 | 98.2±0.3 | 91.0± 0.5 | 98.9±0.1 |
| T2T | 88.2±0.7 | 75.5±0.5 | 89.0±1.0 | 77.2±0.2 | 90.3±0.5 | 82.3±0.2 |
| OpenMatch | 89.6±0.9 | 99.3±0.3 | 92.9±0.5 | **99.7**±0.2 | 94.1±0.5 | 99.3±0.2 |
| POT | 92.1±0.2 | **99.7**±0.1 | 92.9±0.2 | 99.5±0.1 | 93.6±0.1 | 99.4±0.1 |
| SDM | **92.5**±0.4 | 99.6±0.2 | **93.1**±0.1 | 99.5±0.2 | **94.2**±0.1 | **99.4**±0.3 |

Table 2: Top-1 accuracy and AUROC on CIFAR-100 benchmarks under different settings.To verify the statistical significance of the results, we report both the mean and standard deviation of 5 experiments.

| # of Known | 55 | | | | 80 | | | |
|---|---|---|---|---|---|---|---|---|
| # of Labeled | 50 | | 100 | | 50 | | 100 | |
| Metric | Acc | AUROC | Acc | AUROC | Acc | AUROC | Acc | AUROC |
| MTCF | 66.5±1.2 | 81.2±3.4 | 72.1±0.5 | 80.7±4.6 | 59.9±0.8 | 79.4±2.5 | 66.4±0.3 | 73.2±3.5 |
| T2T | 72.2±1.4 | 60.4±1.6 | 73.1±0.8 | 59.8±1.4 | 63.5±1.2 | 55.0±1.8 | 66.8±0.7 | 55.4±1.5 |
| OpenMatch | 72.3±0.4 | 87.0±1.1 | 75.9±0.6 | 86.5±2.1 | 66.6±0.2 | 86.2±0.6 | 70.5±0.3 | 86.8±1.4 |
| POT | 78.7±0.1 | 88.4±0.1 | 81.1±0.1 | 89.5±0.3 | 75.4±0.1 | 88.1±0.3 | 78.1±0.1 | 88.0±0.1 |
| SDM | **78.9**± 0.3 | **91.7**±0.3 | **81.3**±0.4 | **91.3**±0.3 | **75.7**±0.2 | **90.9**±0.3 | **78.3**±0.5 | **90.9**±0.3 |

models need to detect OOD samples exist in $\mathbf{X}_u$. Following the setup in (Sohn et al., 2020) and (Ren et al., 2024), the complete training set is used as unlabeled data, while labeled data is randomly extracted from the training set and the number of samples for each known class is set to different values corresponding to different classification difficulties. We also evaluate SDM on the more challenging ImageNet-30 (Hendrycks & Gimpel, 2017) dataset, where 20 classes are selected as the known classes, with the remaining as the unknown classes.

**Baselines.** To demonstrate the superiority of our method on OOD detection benchmark, we compare our proposed SDM with comprehensive baselines, including Fixmatch(Sohn et al., 2020), MTCF(Yu et al., 2020a), T2T(Huang et al., 2021a), OpenMatch(Saito et al., 2021), and POT(Ren et al., 2024). Since the semi-supervised classification module of SDM in this paper adopts the Fixmatch structure, we consider it as a baseline. The maximum softmax prediction probability as the score function is used to give Fixmatch the ability to perform OOD detection during testing. MTCF and T2T are typical purely neural networks based methods. OpenMatch adopts a method based on OVA classifiers. In addition, to prove the effectiveness of DSA, we also compare SDM with other OT-based method.

**Metrics.** To measure the OOD detection results, we calculate the area under the receiver operating characteristic curve (AUROC) to observe the ability to identify OOD samples. In this paper, we focus on image classification tasks in the presence of OOD samples, so we also report closed-set classification accuracy. In scenarios where we focus on comparing the performance of different methods, we report top-1 accuracy and AUROC, and in scenarios where we focus on analyzing the change with different architectures or parameter settings, we report top-1 to top-5 accuracy to facilitate comprehensive analysis. More settings and implementation details are provided in Section A.5.

## 4.2 RESULTS AND DISCUSSION

The experimental results on CIFAR-10, CIFAR-100, and ImageNet-30 are summarized in Table 1, Table 2, and Table 3. Our analysis reveals a clear performance hierarchy. Methods relying solely on neural networks perform poorly across all datasets, lagging significantly behind proxy-based approaches. While OpenMatch shows competitive results on CIFAR-10, it falls short of OT-based methods, particularly on CIFAR-100 where its closed-set accuracy is lower. Overall, SDM achieves

Table 3: Top-1 accuracy and AUROC on ImageNet-30 benchmarks. 20 classes are selected as known classes, and the remaining classes are unknown classes.

| Method | FixMatch | MTCF | T2T | OpenMatch | POT | Ours |
|---|---|---|---|---|---|---|
| Top1Acc | 91.7±0.5 | 86.4±0.7 | 87.8±0.9 | 89.6±1.0 | 92.0±0.3 | **92.1**±0.4 |
| AUROC | 45.1±1.2 | 93.8±0.8 | 55.7±10.8 | 96.4±0.7 | 97.4±0.4 | **97.7**±0.2 |

Table 4: Ablation study on CIFAR-10 benchmark. Fixmatch-OOD use the maximum softmax prediction probability to give Fixmatch the ability to perform OOD detection during training phase.

| # of Labeled Classes | 50 | | 100 | | 400 | |
|---|---|---|---|---|---|---|
| Metric | Acc | AUROC | Acc | AUROC | Acc | AUROC |
| FixMatch-OOD | 91.7±1.1 | 37.7±0.6 | 92.9±0.7 | 39.8±0.3 | 93.4±0.3 | 40.9±0.6 |
| Fixmatch+DSA | 92.2±0.3 | 99.6±0.1 | 93.0±0.3 | 99.5±0.2 | 94.1±0.2 | 99.4±0.1 |
| Fixmatch+DSA+MCR | **92.5**±0.4 | **99.6**±0.2 | **93.1**±0.1 | **99.5**±0.2 | **94.2**±0.1 | **99.4**±0.3 |

better performance than POT and maintains competitive performance on the more challenging ImageNet-30 dataset with 224×224 resolution. The most significant advantage of SDM is observed in AUROC on CIFAR-100, a gain potentially attributable to the larger number of classes, which causes other methods to not be able to perform OOD detection well.

Table 5: Adaptation to different OT solvers. We report top-1 to top-5 accuracy and AUROC on CIFAR-100 with 55 known classes and 50 labeled samples per known class, with POT as the baseline.

| Metirc | Solver | Type | Solution | Acc | | | | | AUROC |
|---|---|---|---|---|---|---|---|---|---|
| | | | | Top1 | Top2 | Top3 | Top4 | Top5 | |
| POT | POT | Partial OT | Approximate | 78.4 | 87.7 | 91.5 | 93.8 | 95.0 | 90.1 |
| DSA(+EPW) | EPW | OT | Approximate | 79.0 | **88.1** | 91.8 | 93.7 | 95.0 | 91.0 |
| DSA(+Sinkhorn) | Sinkhorn | OT | Approximate | 79.2 | 88.0 | **91.9** | **93.8** | **95.3** | 91.2 |
| DSA(+EMD) | EMD | OT | Exact | **79.5** | 88.0 | 91.8 | 93.7 | 95.2 | **91.5** |

## 4.3 ADAPTATION TO DIFFERENT OT SOLVERS

Since DSA dynamically converts the SemiUOT from $\alpha$ to $\beta$ in each batch into classical OT during training, in theory any classical OT solver can be connected in series behind DSA. In order to verify the adaptability of SDM to different OT solvers, we insert several OT solvers into SDM for trial, including EMD, Sinkhorn and entropic partial wasserstein (EPW) (Cuturi, 2013). Besides, we also report the results of POT as a baseline in Table 5. The SDM plugged into three different OT solvers all outperformed POT in terms of metrics. Interestingly, when we directly connect modules similar to those in POT in series to DSA, we can also observe performance improvements. Since EMD obtains an exact solution, while others obtain approximate solutions, the OOD labels obtained based on DSA and EMD are more reliable.

## 4.4 ANALYSIS AND ABLATION STUDY

**1) Ablation study.** Table 4 reports the overall performance of Fixmath-OOD, Fixmatch+DSA and Fixmatch+DSA+MCR. Fixmatch itself cannot detect OOD samples accurately enough, even when performing OOD detection based on predicted probabilities during inference. When training OOD detector under the supervision of DSA, the AUROC metric which indicates the open-ser OOD detection quality reach an impressively higher lever compared to Fixmatch-OOD. With the support of MCR module as auxiliary task, we observe the improvement of the top-1 closed-set classification accuracy without the sacrifice of AUROC. Performance improvement of the DSA method and the MCR module does not come at the cost of testing latency because only Fixmatch and neural networks

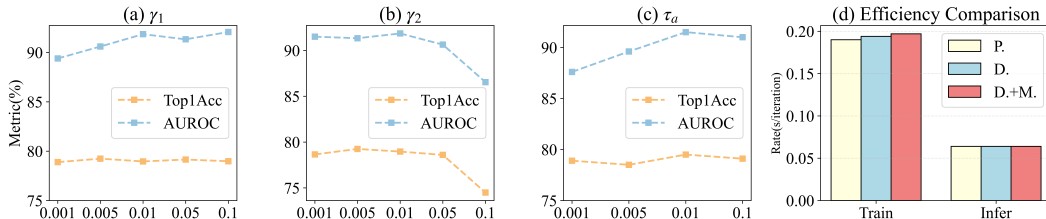

Figure 2: (a) The change of performance with $\gamma_1$ when $\gamma_2$ is fixed to 0.01. (b) The change of performance with $\gamma_2$ when $\gamma_1$ is fixed to 0.01. (c) The change of performance with $\tau_a$. (d) The efficiency comparison of P.(POT), D.(DSA), and D.+M.(DSA+MCR). All results are on CIFAR-100 benchmark with 55 known classes and 50 labeled samples per known class.

that output ID probability are activated during the training phase. With MCR, we can observe an improvement in closed-set accuracy or OOD detection quality without sacrificing the other metric.

**2) Quantitative analysis of the pseudo labels.** In Table 6, We compare the AUROC of the pseudo lables produced by DSA to that of POT with the additional parameter set to 1.25 and 2.5 respectively. All the experiments are conducted on test datasets with labeled training samples as the target samples. The pseudo labels produced by DSA are more accurate than POT.

**3) Influence of $\mathcal{L}_{ood}$ and $\mathcal{L}_{mcr}$.** Since SDM is a multi-task learning framework, the coefficients of the loss functions corresponding to different tasks are important. Figure 2 (a) and (b) depict the changes of top-1 classification accuracy and OOD detection quality with $\gamma_1$ and $\gamma_2$ corresponding to $\mathcal{L}_{ood}$ and $\mathcal{L}_{mcr}$ respectively. Besides, when the value of $\gamma_1$ is too small, such as 0.01, the impact of $\mathcal{L}_{ood}$ on the overall loss is too small to make the quality of OOD detection worse. In other cases, SDM is stable enough to the changes of relevant core parameters.

Table 6: Quantitative analysis of the accuracy of the pseudo labels on CIFAR-100.

| # of Known | 55 | | 80 | |
|---|---|---|---|---|
| # of Labeled | 50 | 100 | 50 | 100 |
| POT($k = 1.25$) | 54.9 | 54.9 | 58.0 | 55.2 |
| POT($k = 2.5$) | 59.2 | 58.7 | 60.0 | 57.9 |
| DSA | **59.7** | **59.2** | **60.1** | **58.3** |

**4) Influence of $\tau_a$.** As we can see that, the larger value of $\tau_a$ in Equation (2) involves more data to get matched. Figure 2 (c) reports the results with $\tau_a$ changes to check the influence of this property on SDM. Given that $\tau_a$ is essentially a scaling factor, our experiments span a fairly large range from 0.001 to 1, where no significant increase or decrease in performance is observed. Thus $\tau_a$ does not need to be overly fine-tuned to adapt to specific tasks and shows sufficient robustness.

**5) Efficiency Analysis.** To validate that the improvement of performance does not come at the cost of significant delay, we count the execution rate of SDM and its variants (i.e., FixMatch+DSA) compared with the baseline. The training rate and testing rate are calculated as the average rate within 1024 iterations in Figure 2 (d). Owing to the acceleration approach of Proposition 3.2, the training latency of DSA is modest compared to POT. The additional computational cost of MCR is also acceptable. Consistent with the analysis in this paper, both DSA and MCR cause no test delay. In addition, the training rate of SDM is 1.25s/iteration on the ImageNet-30 dataset, which means only $6.3\times$ latency at $49\times$ resolution compared to the case on CIFAR-10/100.

## 5 CONCLUSION

In summary, we propose a novel open-set semi-supervised OOD detection framework SDM, which synergistically leverages the advantages of SSL, self-supervised learning, and OT. Through theoretical derivation, we innovatively propose the accelerated DSA method that dynamically converts SemiUOT into classical OT and provides reliable pseudo OOD lables for supervision. Using the MCR module as an auxiliary task for semi-supervised classification and OOD detection, SDM comprehensively surpasses all baselines on all benchmarks, especially dominating the open-set OOD detection benchmarks based on CIFAR-100. Our proposal will further motivate future work to pursue more novel and efficient learning paradigms or frameworks for open-set SSL.

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

## A APPENDIX

### A.1 ALGORITHM OF SDM

---

**Algorithm 2** Open-Set Out-of-Distribution Detection

---

**Require**: Data loader, weak augmentation method $A$, strong augmentation method $\mathcal{A}$, encoder network $f(\cdot)$, closed-set classification head $g(\cdot)$, OOD detection head $h(\cdot)$ and SSL head $\mathcal{G}(\cdot)$.

**for** $\mathbf{X}_l, \mathbf{X}_u$ *in loader* **do**

   **Augmentation:**
   $\mathbf{X}_w, \mathbf{X}_s^1, \mathbf{X}_s^2 = A(\mathbf{X}_u), \mathcal{A}(\mathbf{X}_u), \mathcal{A}(\mathbf{X}_u)$.
   **Encoding:**
   $\mathbf{Z}_l, \mathbf{Z}_w, \mathbf{Z}_s^1, \mathbf{Z}_s^2 = f(\mathbf{X}_l), f(\mathbf{X}_w), f(\mathbf{X}_s^1), f(\mathbf{X}_s^2)$.
   **OOD detection:**
   Compute the cosine similarity $C$ between $\mathbf{Z}_l$ and $\mathbf{Z}_s^1$.
   Obtain OOD pseudo labels $\mathbf{S}'$ via the DSA approach.
   Obtain OOD predicted score $\hat{\mathbf{S}} = h(\mathbf{Z}_l, \mathbf{Z}_s^1)$.
   Compute OOD loss $\mathcal{L}_{ood}$.
   **MCR:**
   Obtain $\mathcal{L}_{mcr}$ via Algorithm 3.
   **Self-supervised classification:**
   Obatin classification probability
   $\mathcal{Y}_l, \mathcal{Y}_w, \mathcal{Y}_s^1 = g(\mathbf{Z}_l), g(\mathbf{Z}_w), g(\mathbf{Z}_s^1)$.
   Compute supervised loss $\mathcal{L}_x$ and unspervised loss $\mathcal{L}_u$.
   **Total loss:**
   $\mathcal{L} = \mathcal{L}_x + \mathcal{L}_u + \gamma_1 \mathcal{L}_{ood} + \gamma_2 \mathcal{L}_{mcr}$.
   Back propagation.

**end**

---

### A.2 ALGORITHM OF MCR

---

**Algorithm 3** Matrix Contrastive Regularization

---

**Input:** Batch of unlabeled sample $\mathbf{X}_u$, strong augmentation method $\mathcal{A}$, encoder network $f(\cdot)$ and contrastive learning head $\mathcal{G}(\cdot)$.

**Output:** MCR loss.

Augmentation: $\mathbf{X}_s^1, \mathbf{X}_s^1 = \mathcal{A}(\mathbf{X}_u), \mathcal{A}(\mathbf{X}_u)$.

Encoding: $\mathbf{Z}_s^1, \mathbf{Z}_s^2 = f(\mathbf{X}_s^1), f(\mathbf{X}_s^2)$.

Projecting and predicting: $\mathcal{Z}_s^1, \mathcal{Z}_s^2 = \mathcal{G}(\mathbf{Z}_s^1), \mathcal{G}(\mathbf{Z}_s^2)$.

Obtain $\mathcal{L}_{\text{Matrix-Alignment}}$ and $\mathcal{L}_{\text{Matrix-Uniformity}}$ via Equation (12).

MCR loss: $\mathcal{L}_{mcr} = \mathcal{L}_{\text{Matrix-Alignment}} + \mathcal{L}_{\text{Matrix-Uniformity}}$.

**Return** $\mathcal{L}_{mcr}$.

---

### A.3 PROOF OF PROPOSITION 3.1

We can rewrite the SemiUOT problem as below:

$$\min_{\boldsymbol{\pi} \geq 0} J = \langle \boldsymbol{C}, \boldsymbol{\pi} \rangle + \tau_a \text{KL}\left(\boldsymbol{\pi}\mathbf{1}_N \| \boldsymbol{a}\right)$$

$$s.t. \text{ (Constraint)} : \boldsymbol{\pi}^\top \mathbf{1}_M = \boldsymbol{b}, \text{(Optional)} : \boldsymbol{\pi}\mathbf{1}_N = \boldsymbol{\alpha}.$$

Note that we do not need to know the exact value of $\boldsymbol{\alpha}$ beforehand. We adopt this optional constraint only for simplifying the following deduction. The Lagrange multipliers of Semi-UOT with KL-Divergence is given as:

$$\max_{\boldsymbol{s} \geq 0, \boldsymbol{u}, \boldsymbol{v}, \zeta} \min_{\boldsymbol{\pi} \geq 0} \mathcal{J} = \tau_a \text{KL}\left(\boldsymbol{\pi}\mathbf{1}_N \| \boldsymbol{a}\right) + \langle \boldsymbol{u} + \zeta, \boldsymbol{\pi}\mathbf{1}_N \rangle + \langle \boldsymbol{v} - \zeta, \boldsymbol{b} \rangle + \mathscr{C}_{\text{SUOT}},$$

where $\mathscr{C}_{\mathrm{SUOT}} = \sum_{i,j}(C_{ij} - u_i - v_j - s_{ij})\pi_{ij} = \langle \boldsymbol{C} - \boldsymbol{u} \otimes \mathbf{1}_N^\top - \mathbf{1}_M \otimes \boldsymbol{v}^\top - \boldsymbol{s}, \boldsymbol{\pi} \rangle$ and $\boldsymbol{u}$, $\boldsymbol{v}$ and $\zeta$ are dual variables. By taking the differentiation on $\pi_{ij}$ we have:

$$\frac{\partial \mathcal{J}}{\partial \pi_{ij}} = \left[ \tau_a \log \frac{\sum_{j=1}^N \pi_{ij}}{a_i} + u_i + \zeta \right] + (C_{ij} - u_i - v_j - s_{ij})$$

$$= C_{ij} + \tau_a \log \frac{\sum_{j=1}^N \pi_{ij}}{a_i} + \zeta - v_j - s_{ij} = 0.$$

Then we can obtain the results:

$$\begin{cases} \sum_{j=1}^N \pi_{ij} = a_i \exp\left(-\frac{u_i + \zeta}{\tau_a}\right) \\ \sum_{i=1}^M \pi_{ij} = b_j \end{cases} \Rightarrow C_{ij} - u_i - v_j - s_{ij} = 0, \quad s_{ij} \geq 0.$$

Thus SemiUOT can be can be regarded as classic optimal transport problem:

$$\min_{\boldsymbol{\pi} \geq 0} \mathcal{J}_{\mathrm{S}} = \langle \boldsymbol{C}, \boldsymbol{\pi} \rangle$$

$$s.t. \boldsymbol{\pi} \mathbf{1}_N = \boldsymbol{a} \odot \exp\left(-\frac{\boldsymbol{u}^* + \zeta^*}{\tau_a}\right), \boldsymbol{\pi}^\top \mathbf{1}_M = \boldsymbol{b}.$$

A.4   PROOF OF PROPOSITION 3.2

We first review the Exact SemiUOT Equation:

$$\min_{\boldsymbol{u}, \zeta} \mathcal{J}_{\mathrm{S}} = \tau_a \sum_{i=1}^M a_i \exp\left(-\frac{u_i + \zeta}{\tau_a}\right) - \sum_{j=1}^N \left[ \inf_{k \in [M]} [C_{kj} - u_k] - \zeta \right] b_j.$$

Then we take the differentiation on $u_i$ to obtain:

$$\left\| \frac{\partial \mathcal{J}_{\mathrm{S}}}{\partial u_i} \right\| = \left\| -a_i \exp\left(-\frac{u_i + \zeta}{\tau_a}\right) + \sum_{j=1}^N \delta\left(i = \arg\min_{k \in [M]} [C_{kj} - u_k]\right) b_j \right\|$$

$$\leq \left\| a_i \exp\left(-\frac{u_i + \zeta}{\tau_a}\right) \right\| + \left\| \sum_{j=1}^N \delta\left(i = \arg\min_{k \in [M]} [C_{kj} - u_k]\right) b_j \right\|$$

$$\leq \left\| \sum_{j=1}^N b_j \right\| + \left\| \sum_{j=1}^N b_j \right\|$$

$$= 2 \sum_{j=1}^N b_j = \mathscr{L}.$$

Therefore, it satisfies the Lipchitz constraints in gradient descend.

$$\|\mathcal{J}_{\mathrm{S}}(\boldsymbol{u}_y) - \mathcal{J}_{\mathrm{S}}(\boldsymbol{u}_x)\| \leq \mathscr{L} \|\boldsymbol{u}_y - \boldsymbol{u}_x\|.$$

Finally we can adopt SGD with step-size as $\eta = \frac{1}{\mathscr{L}}$ for the optimization:

$$u_i^{(\mathrm{new})} = u_i^{(\mathrm{old})} - \frac{1}{\mathscr{L}} \left[ \sum_{j=1}^N \delta\left(i = \arg\min_{k \in [M]} \left[C_{kj} - u_k^{(\mathrm{old})}\right]\right) b_j - a_i \exp\left(-\frac{u_i^{(\mathrm{old})} + \zeta}{\tau_a}\right) \right].$$

Although $\mathcal{J}_{\mathrm{S}}$ is convex and has unique solutions, the presence of $\inf(\cdot)$ renders it a non-smooth function, leading to inefficient optimization. To further accelerate the optimization process, we consider to make a smooth approximation on replacing $\inf(\cdot)$ as $\inf_{k \in [M]}[C_{kj} - f_k] \approx -\epsilon \log[\sum_{k=1}^M e^{\frac{f_k - C_{kj}}{\epsilon}}]$. Note that $\epsilon > 0$ denotes the balanced hyper parameters among the accuracy and function smoothness. Smaller $\epsilon$ (e.g., $\epsilon$ approaches to 0) could lead to more accurate while less smooth solutions. Then

we can obtain the proposed *Approximate SemiUOT Equation* as $\widehat{\mathcal{J}}_S$ by replacing $\inf(\cdot)$ with the smoothness term for $\widehat{f}$ as below:

$$\min_{\boldsymbol{u},\zeta} \widehat{\mathcal{J}}_S = \tau_a \exp\left(-\frac{\zeta}{\tau_a}\right) \sum_{i=1}^{M} a_i \exp\left(-\frac{u_i}{\tau_a}\right) + \sum_{j=1}^{N} \left[\epsilon \log\left[\sum_{k=1}^{M} \exp\left(\frac{u_k - C_{kj}}{\epsilon}\right)\right] + \zeta\right] b_j.$$

Take the differentiation on $u_i$ we can obtain:

$$\frac{\partial \widehat{\mathcal{J}}_S}{\partial u_i} = -a_i \exp\left(-\frac{\zeta}{\tau_a}\right) \exp\left(-\frac{u_i}{\tau_a}\right) + \exp\left(\frac{u_i}{\epsilon}\right) \sum_{j=1}^{N} \left[\frac{\exp\left(-\frac{C_{ij}}{\epsilon}\right)}{\sum_{k=1}^{M} \exp\left(\frac{u_k - C_{kj}}{\epsilon}\right)}\right] b_j = 0.$$

To solve the above problem, we can obtain:

$$u_i^{(l+1)} = \frac{\tau_a \epsilon}{\tau_a + \epsilon} \log\left(a_i \exp\left(-\frac{\zeta}{\tau_a}\right)\right) - \frac{\tau_a \epsilon}{\tau_a + \epsilon} \log\left[\sum_{j=1}^{N} \left[\frac{\exp\left(-\frac{C_{ij}}{\epsilon}\right)}{\sum_{k=1}^{M} \exp\left(\frac{u_k^{(l)} - C_{kj}}{\epsilon}\right)}\right] b_j\right]$$

$$= \mathcal{T}(u_i^{(l)}).$$

We can adopt Banach theorem to verify the convergence of the algorithm.

$$\frac{\partial \mathcal{T}(u_i^{(l)})}{\partial u_i^{(l)}} = -\frac{\tau_a \epsilon}{\tau_a + \epsilon} \frac{\frac{\partial}{\partial u_i^{(l)}}\left(\sum_{j=1}^{N}\left[\frac{\exp\left(-\frac{C_{ij}}{\epsilon}\right)}{\sum_{k=1}^{M} \exp\left(\frac{u_k^{(l)} - C_{kj}}{\epsilon}\right)}\right] b_j\right)}{\sum_{j=1}^{N}\left[\frac{\exp\left(-\frac{C_{ij}}{\epsilon}\right)}{\sum_{k=1}^{M} \exp\left(\frac{u_k^{(l)} - C_{kj}}{\epsilon}\right)}\right] b_j}$$

$$= \frac{\tau_a}{\tau_a + \epsilon} \underbrace{\frac{\sum_{j=1}^{N}\left[\frac{b_j \exp\left(-\frac{C_{ij}}{\epsilon}\right)}{\sum_{k=1}^{M} \exp\left(\frac{u_k^{(l)} - C_{kj}}{\epsilon}\right)} \cdot \frac{\exp\left(\frac{u_i^{(l)} - C_{ij}}{\epsilon}\right)}{\sum_{k=1}^{M} \exp\left(\frac{u_k^{(l)} - C_{kj}}{\epsilon}\right)}\right]}{\sum_{j=1}^{N}\left[\frac{\exp\left(-\frac{C_{ij}}{\epsilon}\right)}{\sum_{k=1}^{M} \exp\left(\frac{u_k^{(l)} - C_{kj}}{\epsilon}\right)}\right] b_j}}_{\leq 1}$$

$$\leq 1.$$

## A.5 IMPLEMENTATION DETAILS

To be fair, we adopt Wide ResNet (WRN) (Zagoruyko & Komodakis, 2016) as the backbone, with WRN-28-2 on CIFAR-10 and WRN-28-8 on CIFAR-100, consistent with the baseline. For each batch, the size of $\mathbf{X}_u$ is twice that of $\mathbf{X}_l$. Since Fixmatch is used in the closed-set semi-supervised classification module of SDM in the experimental part, we set the temperature parameter of pseudo-label to 1 and the threshold of pseudo-label to 0.95. The model is trained with a Nesterov SGD optimizer with 0.9 momentum and $5 \times 10^{-4}$ weight decay. We implement the cosine annealing learning rate adjustment strategy and set the initial learning rate as 0.03. $\tau_a$ in Equation (2) is set to 0.01 when comparing SDM with baselines. The selection of hyperparameters in MCR refers to (Zhang et al., 2023). Putting all the components together, we set the balance factor of $L_x$ and $L_u$ as 1 while setting that of $L_{ood}$ and $L_{mcr}$ as 0.01. For the ImageNet-30 dataset, We use ResNet-18 (He et al., 2016) as the backbone, the experimental settings are the same as the case on CIFAR-10/100.

## A.6 LIMITATIONS

The proposed method has two main limitations as follows:

**1)** Tough the DSA aprroach does not lead to extra testing latency, the computation of OT or SemiUOT in high-dimensional feature embedding is relatively high in training phase, placing demands on the performance of computing devices. In future work, we will focus on further accelerating it via parallel computing strategy and exploring dimension reduction to low-dimensional space for calculation.

**2)** The pseudo labels obtained during training phase is only used for the supervision of neural networks to perform OOD detection. More efforts can be made to enhance the performance of closed-set classification module via the pseudo labels (e.g., through a joint optimization framework).

