# OpenReview forum: "Synergizing Dynamic Score Aggregation with Contrastive Regularization for Open-Set Semi-Supervised Out-of-Distribution Detection"
_ICLR.cc/2026/Conference — ICLR 2026 Conference Withdrawn Submission_

### Official Review · Reviewer_V52c · 2025-10-26

**Soundness:** 2
**Presentation:** 2
**Contribution:** 2
**Rating:** 4
**Confidence:** 3

**Summary:**

The paper introduces a Dynamic Score Aggregation (DSA) module that reformulates out-of-distribution (OOD) detection as a Semi-Unbalanced Optimal Transport (SemiUOT) problem.

**Strengths:**

1. The experiments are comprehensive and convincing, covering CIFAR-10, CIFAR-100, and ImageNet-30 datasets under different open-set SSL settings.
2. The motivation is well-articulated and rooted in a realistic challenge: open-set semi-supervised learning, where unlabeled data may include unknown classes.

**Weaknesses:**

1. There is limited qualitative or visualization analysis of feature distributions or OOD score behavior, which could better illustrate the effect of MCR.
2. No direct runtime comparison against state-of-the-art contrastive SSL baselines is provided; this limits understanding of the real efficiency gains.
3. The integration between DSA and MCR is described at a conceptual level; a deeper analysis of their synergy or mutual effect would enhance understanding.
4. Typographical issues: 'lable' →'label',  'things are often trivial' → 'things are often non-trivial', 'Lipchitz'→ 'Lipschitz', etc.

**Questions:**

See Weaknesses*

---

### Official Review · Reviewer_ewq6 · 2025-10-28

**Soundness:** 3
**Presentation:** 3
**Contribution:** 2
**Rating:** 4
**Confidence:** 3

**Summary:**

This paper addresses the problem of open-set semi-supervised OOD detection.
The key contribution lies in introducing a Dynamic Score Aggregation (DSA) module, which formulates OOD detection as a semi-unbalanced optimal transport (SemiUOT) problem.
And the authors further dynamically convert it into a classical optimal transport (OT) problem, with a Matrix Contrastive Regularization (MCR) module that enhances feature discrimination.
Experiments are conducted on CIFAR-10, CIFAR-100, and ImageNet-30 benchmarks, showing the competitive performance compared with existing methods.

**Strengths:**

1. The proposed dynamic score aggregation to estimate pseudo OOD scores seems interesting and potentially effective.

2. The paper presents a compelling synergy between optimal transport and self-supervised learning.

**Weaknesses:**

1. The DSA approximation is designed for efficiency and does not affect testing latency.
However, the inherent complexity of computing optimal transport in high-dimensional feature space remains a non-trivial training overhead.

2. The SemiUOT formulation assumes uniform discrete distributions for the source and target.
The impact of deviations from this assumption in real-world, non-uniform data distributions on the method's performance is not thoroughly discussed.

**Questions:**

1. Could the author clarify the conclusion of Propositions 3.1 and 3.2?
It appears that the statement of the conclusion and its proof are mixed together.
In addition, from which definitions or lemmas are these three propositions derived?

---

### Official Review · Reviewer_7TM9 · 2025-10-31

**Soundness:** 3
**Presentation:** 2
**Contribution:** 2
**Rating:** 4
**Confidence:** 4

**Summary:**

They propose a new approach DSA and MCR modules for an open-set semi-supervised learning task. DSA provides pseudo-labels for the target samples in terms of their OOD score, and aims to convert SemiUOT into a classical optimal transport (OT) formulation, which gives more computationally efficient and accurate results for this task. MCR is applied to source domains and applies contrastive loss.
According to their empirical results, the proposed method shows some improvements over existing approaches and provides insights into their DSA modules.

**Strengths:**

1. Their main contribution is in the DSA module, which gives pseudo-labels to unlabeled target samples for OOD detection. They propose to transform the SemiUOT problem to a classical OT problem to solve it in an efficient and accurate way.

2. Their proposed approach shows small improvements over existing approaches in terms of AUROC.

3. They provide diverse analyses on their DSA modules, e.g., ablation for OT solvers, and hyper-parameters, which can be insightful.

**Weaknesses:**

1. They need improvements in their presentation. For example, Eq. 3 is not clear. h is not introduced at this point, and the motivation of Eq. 3 is not clear. They need improvements in introducing their architectures.

2. According to Table 4, the performance gain from the proposed MCR module is marginal. It appears that the DSA module is the most effective one, and the MCR introduces a very small gain. MCR might not be necessary in this method.

3. MCR loss combines two losses, contrastive loss and their proposed loss. Ablation studies on this loss have not been performed.

4. Line 51-52 mentions that OpenMatch is computationally inefficient. However, I think in terms of training-time complexity, the proposed approach is more inefficient.

5. They seem to require three unique hyperparameters to tune for their approach. Is there any comparison to existing approaches in terms of the number of hyperparameters?

**Questions:**

Please respond to the weaknesses above.

---

### Official Review · Reviewer_MnoM · 2025-10-31

**Soundness:** 3
**Presentation:** 3
**Contribution:** 2
**Rating:** 4
**Confidence:** 3

**Summary:**

This paper proposes a novel open-set OOD detection framework named SDM, which synergizes Dynamic Score Aggregation (DSA) and Matrix Contrastive Regularization (MCR). Experimental results demonstrate the superiority of the proposed model.

**Strengths:**

1) The proposed DSA module dynamically converts the SemiUOT problem into classical optimal transport.

2) The proposed MCR enhances feature discrimination without introducing inference latency.

3) Extensive experimental results on multiple datasets demonstrate that the proposed SDM achieves competitive, and often superior, performance compared to state-of-the-art methods.

**Weaknesses:**

1) Clarify the “dynamic” nature of DSA: the word “dynamic” is pivotal to the method, yet its meaning—how and why the OT cost matrix or constraints are updated during training—remains vague and should be spelled out explicitly.

2) Articulate MCR’s novelty: the paper demonstrates that Matrix Contrastive Regularization works, but it does not clearly differentiate MCR from existing contrastive or self-supervised regularizers; a sharper discussion of what is new (e.g., matrix-level objective, specific sampling strategy, or theoretical property) is needed.

3) Expand baselines: the current experiments omit several recent state-of-the-art OOD-detection methods; including them would strengthen the empirical claim.

4) Justify DSA’s superior pseudo-labels: provide qualitative insight—e.g., side-by-side visualization of transport plans π from DSA vs. standard POT on a toy example—so readers can intuitively see why DSA yields cleaner scores.

5) How does MCR help OOD detection? The claim is that it improves feature discrimination. A visualization of the feature space (e.g., using t-SNE/UMAP) with and without MCR, colored by ID/OOD status, could powerfully illustrate this effect.

**Questions:**

1) The paper correctly lists the high computational cost of OT in the training phase as a limitation. However, a more concrete quantification would be helpful.

2) The comparison baseline methods are not sufficient.

---

### Note · Authors · 2025-11-13

I have read and agree with the venue's withdrawal policy on behalf of myself and my co-authors.